# The MAGICAL Benchmark for Robust Imitation

**Sam Toyer**      **Rohin Shah**      **Andrew Critch**      **Stuart Russell**
Department of Electrical Engineering and Computer Sciences
University of California, Berkeley
{sdt,rohinmshah,critch,russell}@berkeley.edu

## Abstract

Imitation Learning (IL) algorithms are typically evaluated in the same environment
that was used to create demonstrations. This rewards precise reproduction of
demonstrations in one particular environment, but provides little information about
how robustly an algorithm can generalise the demonstrator's intent to substantially
different deployment settings. This paper presents the MAGICAL benchmark suite,
which permits *systematic* evaluation of generalisation by quantifying robustness
to different kinds of distribution shift that an IL algorithm is likely to encounter
in practice. Using the MAGICAL suite, we confirm that existing IL algorithms
overfit significantly to the context in which demonstrations are provided. We also
show that standard methods for reducing overfitting are effective at creating narrow
perceptual invariances, but are not sufficient to enable transfer to contexts that
require substantially different behaviour, which suggests that new approaches will
be needed in order to robustly generalise demonstrator intent. Code and data for
the MAGICAL suite is available at https://github.com/qxcv/magical/.

## 1   Introduction

Imitation Learning (IL) is a practical and accessible way of programming robots to perform useful
tasks [6]. For instance, the owner of a new domestic robot might spend a few hours using tele-
operation to complete various tasks around the home: doing laundry, watering the garden, feeding
their pet salamander, and so on. The robot could learn from these demonstrations to complete the
tasks autonomously. For IL algorithms to be useful, however, they must be able to learn how to
perform tasks from few demonstrations. A domestic robot wouldn't be very helpful if it required
thirty demonstrations before it figured out that you are deliberately washing your purple cravat
separately from your white breeches, or that it's important to drop bloodworms *inside* the salamander
tank rather than next to it. Existing IL algorithms assume that the environment observed at test time
will be identical to the environment observed at training time, and so they cannot generalise to this
degree. Instead, we would like algorithms that solve the task of *robust IL*: given a small number of
demonstrations in one training environment, the algorithm should be able to generalise the intent
behind those demonstrations to (potentially very different) deployment environments.

One barrier to improved algorithms for robust IL is a lack of appropriate benchmarks. IL algorithms
are commonly tested on Reinforcement Learning (RL) benchmark tasks, such as those from OpenAI
Gym [37, 23, 27, 8]. However, the demonstrator intent in these benchmarks is often trivial (e.g. the
goal for most of Gym's MuJoCo tasks is simply to run forward), and limited variation in the initial
state distribution means that algorithms are effectively being evaluated in the same setting that was
used to provide demonstrations. Recent papers on Inverse Reinforcement Learning (IRL)—which
is a form of IL that infers a reward under which the given demonstrations are near-optimal—have
instead used "testing" variants of standard Gym tasks which differ from the original demonstration
environment [17, 39, 32, 33]. For instance, Fu et al. [17] trained an algorithm on demonstrations from
the standard "Ant" task from Gym, then tested on a variant of the task where two of the creature's

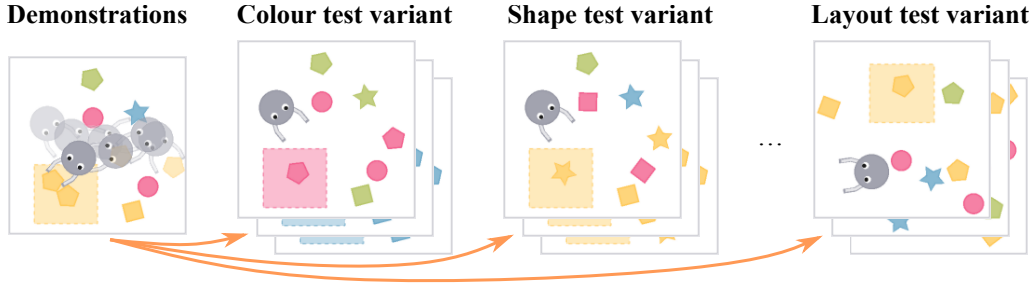

| Demonstrations | Colour test variant | Shape test variant | Layout test variant |

Figure 1: Unlike existing IL benchmarks, MAGICAL makes a distinction between *demonstration* and *test* variants of a task. Demonstrations are all provided in one particular configuration of the world (the "demonstration variant"). The learnt policy (or reward function) is then evaluated across a set of *test variants*, each of which randomise one aspect of the environment, such as block colour or shape, environment layout, dynamics, etc. This makes it possible to understand precisely which aspects of the underlying task the algorithm has been able to infer from demonstrations.

four legs were disabled. Splitting the environment into such "training" and "test" variants makes it possible to measure the degree to which an algorithm overfits to task-irrelevant features of the supplied demonstrations. However, there is so far no standard benchmark for robust IL, and researchers must instead use ad-hoc adaptations of RL benchmarks—such as the modified Ant benchmark and similar alternatives discussed in Section 5—to evaluate intent generalisation.

To address the above issues, we introduce the Multitask Assessment of Generalisation in Imitative Control ALgorithms (MAGICAL). Each MAGICAL task occurs in the same 2D "MAGICAL universe", where environments consist of a robot with a gripper surrounded by a variable number of objects in a fixed-size workspace. Each task is associated with a *demonstration variant*, which is a fixed initial state from which all human demonstrations are provided. A task is also associated with a set of *test variants* for which no demonstrations are provided. As illustrated in Fig. 1, the test variants each randomise a different aspect of the world, such as object colour, transition dynamics, or object count. Randomising attributes of objects and the physics of the world lets us evaluate the ability of a robust IL algorithm to perform *combinatorial generalisation* [5]. For instance, given a demonstration of the robot pushing a red square across the workspace, an algorithm should be able to push a yellow circle across the workspace; given a demonstration of three green and yellow blocks being placed in a line, an algorithm should also be able to place four red and blue blocks in a line; and so on.

MAGICAL has several advantages over evaluation methods for standard (non-robust) IL:

- **No "training on the test set".** Evaluating in the same setting that was used to give demonstrations allows algorithms to exploit features that might not be present during deployment. Having separate test variants for a task allows us to identify this kind of overfitting.
- **Distinguishes between different types of transfer.** Each test variant evaluates robustness to a distinct, semantically meaningful axis of variation. This makes it possible to characterise precisely which aspects of the provided demonstrations a given algorithm is relying on, and to diagnose the causes of over- or under-fitting.
- **Enables knowledge reuse between tasks.** Each MAGICAL task requires similar concepts and low-level skills to solve. Different tasks can therefore provide "background knowledge" for multi-task and meta-IL algorithms, such as knowledge that objects can have different colours, or that objects with different shapes respond in a particular way when grasped.

Our experiments in Section 4 demonstrate the brittleness of standard IL algorithms, particularly under large shifts in object position or colour. We also show that common methods for improving generalisation—such as multitask training, data augmentation, and alternative camera views—sometimes improve robustness to small changes, but still fail to generalise to more extreme ones.

## 2 MAGICAL: Systematically evaluating robust IL

We will now introduce the main elements of the MAGICAL benchmark. We first describe the abstract setup of our benchmark, then detail the specific tasks and variants available in the benchmark.

## 2.1 Benchmark setup

The MAGICAL benchmark consists of a set of *tasks* $\mathcal{T}_1, \mathcal{T}_2, \ldots, \mathcal{T}_m$. Each task can in turn be broken down into *variants* of a single base Markov Decision Process (MDP) that provide different state distributions and "physics" for an agent. Formally, each task $\mathcal{T} = (S, v^D, \mathcal{V})$ consists of a *scoring function* $S(\tau)$, a *demonstration variant* $v^D$, and a set of additional *test variants* $\mathcal{V} = \{v_1, v_2, \ldots, v_n\}$. The scoring function $S(\tau)$ takes a trajectory $\tau = (s_0, a_0, s_1, a_1, \ldots, s_T, a_T)$ and assigns it a score $S(\tau) \in [0, 1]$, where 0 is the score of a no-op policy, and 1 is the score of a perfect demonstrator. Unlike a reward function, $S(\tau)$ need not be Markovian. In order to evaluate generalisation, the variants are split into a single demonstration variant $v^D$ and a set of test variants $\mathcal{V}$.

In our domestic robotics analogy, $v^D$ might represent a single room and time-of-day in which demonstrations for some domestic task collected, while each test variant $v \in \mathcal{V}$ could represent a different room, different time-of-day, and so on. Algorithms are expected to be able to take demonstrations given only in demonstration variant $v^D$, then generalise the intent behind those demonstrations in order to achieve the same goal in each test variant $v \in \mathcal{V}$. This can be viewed either as a form of domain transfer, or as ordinary generalisation using only a single sample from a hypothetical distribution over all possible variants of each task.

Formally, each variant associated with a task $\mathcal{T}$ defines a distribution over reward-free MDPs. Specifically, a variant $v = (p_0, p_\rho, H)$ consists of an *initial state distribution* $p_0(s_0)$, a *dynamics distribution* $p_\rho(\rho)$, and a *horizon* $H$. States are fully observable via an image-based observation space. Further, all variants have the same state space, the same observation space, and the same action space, which we discuss below. In addition to sampling an initial state $s_0 \sim p_0(s_0)$, at the start of each trajectory, a *dynamics vector* $\rho \in \mathbb{R}^d$ is also sampled from the dynamics distribution $p_\rho(\rho)$. Unlike the state, $\rho$ is not observable to the agent; this vector controls aspects of the dynamics such as friction and motor strength. Finally, the horizon $H$ defines a fixed length for all trajectories sampled from the MDP associated with the variant $v$. Each variant associated with a given task has the same fixed horizon $H$ to avoid "leaking" information about the goal through early termination [27].

All tasks and variants in the MAGICAL benchmark share a common continuous state space $\mathcal{S}$. A state $s \in \mathcal{S}$ consists of a configuration (pose, velocity, and gripper state) $q_R$ for the robot, along with object configurations $\mathcal{O} = \{o_1, \ldots, o_E\}$ (where the number of objects in $s_0$ may be random). In addition to pose, each object configuration $o_i$ includes an object *type* and a number of fixed attributes. Objects can be of two types: *blocks* (small shapes that can be pushed around by the agent) and *goal regions* (coloured rectangles that the agent can drive over, but not push around). Each block has a fixed shape (square, pentagon, star, or circle) and colour (red, green, blue, or yellow). Each goal region has a fixed colour, width, and height. In order to facilitate generalisation across tasks with a different number of objects, we use a common image-based observation space and discrete, low-level action space for all tasks, which we describe in detail in Appendix A.1. At an implementation level, we expose each variant of each task as a distinct Gym environment [8], which makes it straightforward to incorporate MAGICAL into existing IL and RL codebases.

## 2.2 Tasks and variants

With the handful of building blocks listed in the previous section, we can create a wide variety of tasks, which we describe in Section 2.2.1. The object-based structure of the environment also makes it easy to evaluate combinatorial generalisation by randomising one or more attributes of each object while keeping the others fixed, as described in Section 2.2.2.

### 2.2.1 Tasks

Tasks in the MAGICAL suite were chosen to balance three desiderata. First, given a handful of trajectories from the demonstration variant of a task, it should be possible for a human observer to infer the goal with sufficient accuracy to solve the test variants. We have chosen demonstration variants (illustrated in Fig. 2) that rule out obvious misinterpretations, like mistakenly identifying colour as being task-relevant when it is not. Second, the tasks should be constructed so that they involve complementary skills that meta- and multi-task learning algorithms can take advantage of. In our tasks, these "shared skills" include block manipulation; identification of colour or shape; and relational reasoning. Third, the demonstration variant of each task must be solvable by existing (non-robust) IL algorithms. This ensures that the main challenge of the MAGICAL suite lies in

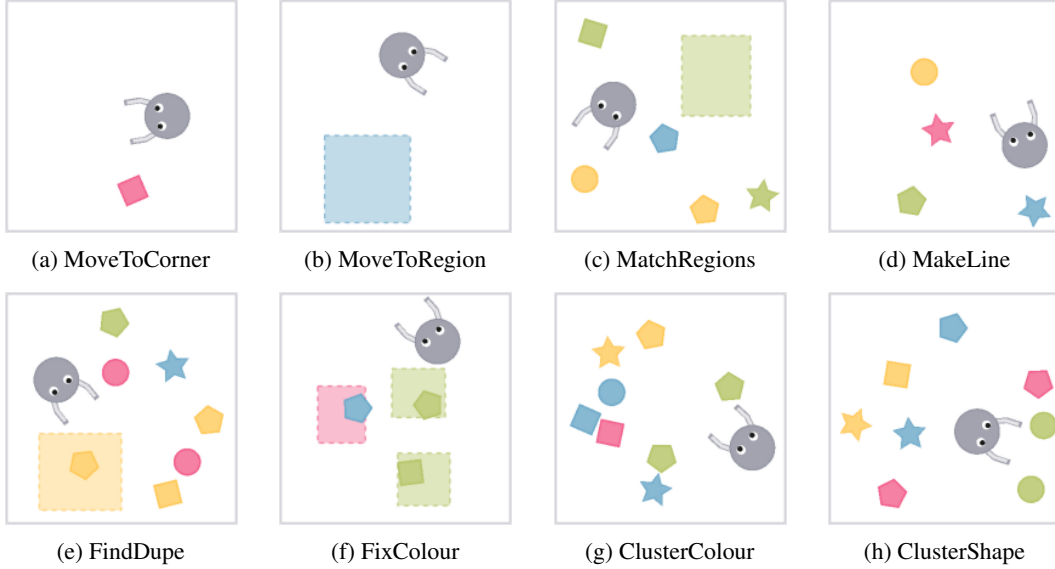

(a) MoveToCorner    (b) MoveToRegion    (c) MatchRegions    (d) MakeLine

(e) FindDupe    (f) FixColour    (g) ClusterColour    (h) ClusterShape

Figure 2: Demonstration variants for MAGICAL tasks. Appendix A shows an example demonstration for each task.

*generalising* to the test variants (robust IL), as opposed to reproducing the demonstrator's behaviour in the demonstration variant (standard IL). This section briefly describes the resulting tasks; detailed discussion of horizons, score functions, etc. is deferred to Appendix A.

**Move to Corner (MTC)**  The robot must push a single block from one corner of the workspace to the diagonally opposite corner. Test variants are constrained so that the robot and block start near the lower right corner. The score is $S(\tau) = 1$ if the block finishes the trajectory in the top left eighth of the workspace, and decreases to zero as the block gets further from the top left corner.

**MoveToRegion (MTR)**  The robot must drive inside a goal region and stay there. There are no blocks in the demonstration or test variants. Further, variants only have one goal region to ensure that the objective is unambiguous. The agent's score is $S(\tau) = 1$ if the robot's body is inside the goal region at the end of the trajectory, and $S(\tau) = 0$ otherwise.

**MatchRegions (MR)**  There is a set of coloured blocks and a goal region visible to the robot, and the robot must push all blocks of the same colour as the goal region into the goal region. Test variants are constrained to have one goal region and at least one block of the same colour as that goal region. A perfect score is given upon termination if the goal regions contains all and only blocks of the goal region's colour, with penalties for excluding any blocks of the goal colour, or including other blocks.

**MakeLine (ML)**  Here the objective is for the robot to arrange all the blocks in the workspace into a single line. A perfect score is given if all blocks are approximately colinear and close together; a penalty is given for each block that does not form part of the longest identifiable line. Refer to Appendix A for details on how a "line" is defined.

**FindDupe (FD)**  Similar to MatchRegions, except the goal region initially contains a "query" block which has the same shape and colour as at least one other block outside the goal region. The objective is to push at least one of those duplicate blocks into the goal region, which yields a perfect score. Penalties are given for knocking the query block out of the goal region, failing to find a duplicate, or pushing non-duplicate blocks into the goal region.

**FixColour (FC)**  In each variant of this task, the workspace contains a set of non-overlapping goal regions. Each goal region contains a single block, and exactly one block in the workspace will have a different colour to its enclosing goal region. A perfect score is given for pushing that block out of its enclosing goal region and into an unoccupied part of the workspace, without disturbing other blocks.

**ClusterColour (CC) and ClusterShape (CS)**  The robot is confronted with a jumble of blocks of different colours and shapes. It must push the blocks into clusters of either uniform colour (in the CC

task), or uniform shape (in the CS task). Test variants are constrained to include at least one block of each colour and each shape. A perfect score is given for creating four spatially distinct clusters corresponding to each of the four colours (CC) or shapes (CS), with a penalty proportional to the number of blocks that do not belong to an identifiable cluster.

### 2.2.2 Test variants

In addition to its demonstration variant, each of the tasks above has a set of associated test variants. Some variants are not supported for tasks that do not have any blocks, or where the initial state is otherwise restricted, as documented in Table 2 of Appendix A.

**Jitter** Takes demo variant and randomly perturbs the poses of the robot and all objects by up to 5% of the maximum possible range. Failure on this variant indicates severe overfitting to the demonstration variant (e.g. by memorising action sequences).

**Layout** Completely randomises the position and orientation of the robot and all blocks, plus position and dimensions of goal regions; a more challenging version of Jitter.

**Colour** Block colours are randomly reassigned as appropriate for the task. This tests whether the agent is responsive to block colour (when it is task-relevant, like in CC and MR), or is correctly ignorant of colour (when it is irrelevant, like in MTC and CS).

**Shape** Similar to Colour, except the shapes of blocks are randomised rather than the colours. This variant either tests for appropriate responsiveness or invariance to shape, depending on whether shape is task-relevant.

**CountPlus** The number of blocks is randomised (along with shape, colour, and position) to test whether the agent can handle "larger" or "smaller" problems (i.e. "generalisation to $n$" [35]).

**Dynamics** Subtly randomises friction of objects and the robot against the workspace, as well as force of robot motors (for rotation, forward/backward motion, and the gripper).

**All** Combines all applicable variants for a task (e.g. Layout, Colour, Shape, CountPlus, Dynamics).

## 3 Data-efficient intent disambiguation

Succeeding at the MAGICAL benchmark requires agents to generalise the intent behind a set of demonstrations to substantially different test variants. We anticipate that resolving the ambiguity inherent in this task will require additional sources of information about the demonstrator's goal beyond just single-task demonstrations. In this section, we review two popular non-robust IL algorithms, as well as some common ways in which alternative sources of goal information are incorporated into these algorithms to improve generalisation.

### 3.1 Baseline methods

Our first baseline method is Behavioural Cloning (BC). BC treats a demonstration dataset $\mathcal{D}$ as an undistinguished collection of state-action pairs $\{(s_1, a_1), \ldots, (s_M, a_M)\}$. It then optimises the parameters $\theta$ of the policy $\pi_\theta(a \mid s)$ via gradient descent on the log loss

$$\mathcal{L}_{\text{bc}}(\theta; \mathcal{D}) = -\mathbb{E}_{\mathcal{D}} \log \pi_\theta(a \mid s) .$$

Our second baseline method is Generative Adversarial IL (GAIL) [23]. GAIL casts IL as a GAN problem [19], where the generator $\pi_\theta(a \mid s)$ is an imitation policy, and the discriminator $D_\psi : \mathcal{S} \times \mathcal{A} \to [0, 1]$ is tasked with distinguishing imitation behaviour from expert behaviour. Specifically, GAIL uses alternating gradient descent to approximate a saddle point of

$$\max_\theta \min_\psi \left\{ \mathcal{L}_{\text{adv}}(\theta, \psi; \mathcal{D}) = -\mathbb{E}_{\pi_\theta} \log D_\psi(s, a) - \mathbb{E}_{\mathcal{D}} \log(1 - D_\psi(s, a)) + \lambda H(\pi_\theta) \right\} ,$$

where $H$ denotes entropy and $\lambda \geq 0$ is a policy regularisation parameter.

We also included a slight variation on GAIL which (approximately) minimises Wasserstein divergence between occupancy measures, rather than Jensen-Shannon divergence. We refer to this baseline as

WGAIL-GP. In analogy with WGAN-GP [20], WGAIL-GP optimises the cost

$$\max_{\theta} \min_{\psi} \left\{ \mathcal{L}_{\text{w-gp}}(\theta, \psi; \mathcal{D}) = \mathbb{E}_{\mathcal{D}} D_\psi(s, a) - \mathbb{E}_{\pi_\theta} D_\psi(s, a) + \lambda_{\text{w-gp}} \mathbb{E}_{\frac{1}{2}\pi_\theta + \frac{1}{2}\mathcal{D}} (\|\nabla_s D(s, a)\|_2 - 1)^2 \right\},$$

The gradient penalty approximately enforces 1-Lipschitzness of the discriminator by encouraging the norm of the gradient to be 1 at points between the support of $\pi_\theta$ and $\mathcal{D}$. Since actions were discrete, we did not enforce 1-Lipschitzness with respect to the action input. We also did not backpropagate gradients with respect to the gradient penalty back into the policy parameters $\theta$, since the gradient penalty is only intended as a soft constraint on $D$.

In addition to these baselines, we also experimented with Apprenticeship Learning (AL). Unfortunately we could not get AL to perform well on most of our tasks, so we defer further discussion of AL to Appendix B.

### 3.2 Using multi-task data

As noted earlier, the MAGICAL benchmark tasks have similar structure, and should in principle benefit from multi-task learning. Specifically, say we are given a multi-task dataset $\mathcal{D}_{\text{mt}} = \{\mathcal{D}(\mathcal{T}_i, v_i^D, n_i)\}_{i=1}^M$, where $\mathcal{D}(\mathcal{T}_i, v, n)$ denotes a dataset of $n$ trajectories for variant $v$ of task $\mathcal{T}_i$. For BC and GAIL, we can decompose the policy for task $\mathcal{T}_i$ as $\pi_\theta^i = g_\theta^i \circ f_\theta$, where $f_\theta : \mathcal{S} \to \mathbb{R}^d$ is a multi-task state encoder, while $g_\theta^i : \mathbb{R}^d \to \Delta(\mathcal{A})$ is a task-specific policy decoder. We can also decompose the GAIL discriminator as $D_\psi^i = s_\psi^i \circ r_\psi$, where $r_\psi : \mathbb{S} \times \mathbb{A} \to \mathbb{R}^d$ is shared and $s_\psi^i : \mathbb{R}^d \to [0, 1]$ is task-specific. We then modify the BC and GAIL objectives to

$$\mathcal{L}_{\text{bc}}(\theta; \mathcal{D}_{\text{mt}}) = \sum_{i=1}^M \mathcal{L}_{\text{bc}}(\theta; \mathcal{D}(\mathcal{T}_i, v_i^D, n_i)) \text{ and } \mathcal{L}_{\text{adv}}(\theta, \psi; \mathcal{D}_{\text{mt}}) = \sum_{i=1}^M \mathcal{L}_{\text{adv}}(\theta, \psi; \mathcal{D}(\mathcal{T}_i, v_i^D, n_i)).$$

### 3.3 Domain-specific priors and biases

Often the most straightforward way to improve the robustness of an IL algorithm is to constrain the solution space to exclude common failure modes. For instance, one could use a featurisation that only captures task-relevant aspects of the state. Such priors and biases are generally domain-specific; for the image-based MAGICAL suite, we investigated two such biases:

- **Data augmentation:** In MAGICAL, our score functions are invariant to whether objects are repositioned or rotated slightly; further, human observers are typically invariant to small changes in colour or local image detail. As such, we used random rotation and translation, Gaussian noise, and colour jitter to augment training data for the BC policy and GAIL discriminator. This can be viewed as a post-hoc form of domain randomisation, which has previously yielded impressive results in robotics and RL [2]. We found that GAIL discriminator augmentations were necessary for the algorithm to solve more-challenging tasks, as previously observed by Zolna et al. [41]. In BC, we found that policy augmentations improved performance on both demonstration and test variants.

- **Ego- and allocentric views:** Except where indicated otherwise, all of the experiments in Section 4 use an *egocentric* perspective, which always places the agent at the same position (and in the same orientation) within the agent's field of view. This contrasts with an *allocentric* perspective, where observations are focused on a fixed region of the environment (in our case, the extent of the workspace), rather than following the agent's position. In the context of language-guided visual navigation, Hill et al. [22] previously found that an egocentric view improved generalisation to unseen instructions or unseen visual objects, despite the fact that it introduces a degree of partial observability to the environment.

## 4   Experiments

Our empirical evaluation has two aims. First, to confirm that single-task IL methods fail to generalise beyond the demonstration variant in the MAGICAL suite. Second, to analyse the ways in which the common modifications discussed in Section 3 affect generalisation.

| Method | Demo | Jitter | Layout | Colour | Shape |
|---|---|---|---|---|---|
| BC (single-task) | 0.64±0.29 | 0.56±0.27 | 0.14±0.16 | 0.39±0.30 | 0.52±0.33 |
| Allocentric | 0.58±0.33 | 0.48±0.29 | 0.04±0.04 | 0.42±0.32 | 0.50±0.37 |
| No augmentations | 0.55±0.37 | 0.37±0.30 | 0.12±0.15 | 0.33±0.30 | 0.41±0.33 |
| No trans./rot. aug. | 0.55±0.37 | 0.41±0.31 | 0.13±0.15 | 0.33±0.30 | 0.43±0.35 |
| Multi-task | 0.59±0.33 | 0.53±0.31 | 0.14±0.18 | 0.30±0.25 | 0.51±0.36 |
| GAIL (single-task) | 0.72±0.35 | 0.69±0.33 | 0.22±0.23 | 0.27±0.24 | 0.60±0.42 |
| Allocentric | 0.57±0.46 | 0.49±0.40 | 0.03±0.03 | 0.39±0.36 | 0.50±0.45 |
| No augmentations | 0.44±0.42 | 0.32±0.31 | 0.09±0.12 | 0.19±0.23 | 0.28±0.33 |
| WGAIL-GP | 0.42±0.38 | 0.33±0.32 | 0.14±0.20 | 0.10±0.11 | 0.33±0.33 |
| Multi-task | 0.37±0.41 | 0.33±0.36 | 0.16±0.25 | 0.11±0.12 | 0.28±0.36 |

Table 1: Score statistics for a subset of variants and compared algorithms. We report the mean and standard deviation of test scores aggregated across *all* tasks, with five seeds per algorithm and task. Darker colours indicate higher scores.

## 4.1 Experiment details

We evaluated all the single- and multi-task algorithms in Section 3, plus augmentation and perspective ablations, on all tasks and variants. Each algorithm was trained five times on each task with different random seeds. In each run, the training dataset for each task consisted of 10 trajectories from the demo variant. All policies, value functions, and discriminators were represented by Convolutional Neural Networks (CNNs). Observations were preprocessed by stacking four temporally adjacent RGB frames and resizing them to 96×96 pixels. For multi-task experiments, task-specific weights were used for the final fully-connected layer of each policy/value/discriminator network, but weights of all preceding layers were shared. The BC policy and GAIL discriminator both used translation, rotation, colour jitter, and Gaussian noise augmentations by default. The GAIL policy and value function did not use augmented data, which we found made training unstable. Complete hyperparameters and data collection details are listed in Appendix B. The IL algorithm implementations that we used to generate these results are available on GitHub,[1] as is the MAGICAL benchmark suite and all demonstration data.[2]

## 4.2 Discussion

Due to space limitations, this section addresses only a selection of salient patterns in the results. Table 1 provides score statistics for a subset of algorithms and variants, averaged across *all* tasks. See Section 2.2.1 for task name abbreviations (MTR, FC, etc.). Because the tasks vary in difficulty, pooling across all tasks yields high score variance in Table 1. Actual score variance for each method is much lower when results are constrained to just one task; refer to Appendix C for complete results.

**Overfitting to position** All algorithms exhibited severe overfitting to the *position* of objects. The Layout, CountPlus, and All variants yielded near-zero scores in all tasks except MTC and MTR, and on many tasks there was also poor transfer to the Jitter variant. For some tasks, we found that the agent would simply execute the same motion regardless of its initial location or the positions of task-relevant objects. This was true on the FC task, where the agent would always execute a similar forward arc regardless of its initial position, and also noticeable on MTC and FD, where the agent would sometimes move to the side of a desired block when it was shifted slightly. For BC, this issue was ameliorated by the use of translation and rotation augmentations, presumably because the policy could better handle small deviations from the motions seen at training time.

**Colour and shape transfer** Surprisingly, BC and GAIL both struggled with colour transfer to a greater degree than shape transfer on several tasks, as evidenced by the aggregated statistics for Colour and Shape variants in Table 1. Common failure modes included freezing in place or moving in the wrong direction when confronted with an object of a different colour to that seen at training time. In contrast, in most tasks where shape invariance was desirable (including MTC, MR, ML, and FC), the agent had no trouble reaching and manipulating blocks of different shapes. Although colour

jitter was one of the default augmentations, the BC ablations in Table 1 suggest that almost all of the advantage of augmentations comes from the use of translation/rotation augmentations. In particular, we did not find that colour jitter greatly improved performance on tasks where the optimal policy was colour-invariant. In spite of exposing the networks to a greater range of colours at train time, multitask training also failed to improve colour transfer, as we discuss below. Although translation and rotation sometimes improved colour transfer (e.g. for BC on FindDupe in Table 7), it is not clear why this was the case. We speculate that these augmentations could have encouraged the policy to acquire more robust early-layer features for edge and corner detection that did not rely on just one colour channel.

**Multi-task transfer** Plain multi-task learning had mixed effects on generalisation. In some cases it improved generalisation (e.g. for BC on FC), but in most cases it led to unchanged or *negative* transfer, as in the Colour test variants for MTC, MR, and FD. This could have been because the policy was using colour to distinguish between tasks. More speculatively, it may be that a multi-task BC or GAIL loss is not the best way to incorporate off-task data, and that different kinds of multi-task pretraining are necessary (e.g. learning forward or inverse dynamics [9]).

**Egocentric view and generalisation** The use of an allocentric (rather than egocentric) view did not improve generalisation or demo variant performance for most tasks, and sometimes decreased it. Table 1 shows the greatest performance drop on variants that change object position, such as Layout and Jitter. For example, in MTR we found that egocentric policies tended to rotate in one direction until the goal region was in the centre of the agent's field of view, then moved forward to reach the region, which generalises well to different goal region positions. In contrast, the allocentric policy would often spin in place or get stuck in a corner when confronted with a goal region in a different position. This supports the hypothesis of Hill et al. [22] that the egocentric view improves generalisation by creating positional invariances, and reinforces the value of being able to independently measure generalisation across distinct axes of variation (position, shape, colour, etc.).

## 5    Related work

There are few existing benchmarks that specifically examine robust IL. The most similar benchmarks to MAGICAL have appeared alongside evaluations of IRL and meta-IL algorithms. As noted in Section 1, several past papers employ "test" variants of standard Gym MuJoCo environments to evaluate IRL generalisation [17, 39, 32, 33], but these modified environments tend to have trivial reward functions (e.g. "run forward") and do not easily permit cross-environments transfer. Xu et al. [38] and Gleave and Habryka [18] use gridworld benchmarks to evaluate meta- and multi-task IRL, and both benchmarks draw a distinction between demonstration and execution environments within a meta-testing task. This distinction is similar in spirit to the demonstration/test variant split in MAGICAL, although MAGICAL differs in that it has more complex tasks and the ability to evaluate generalisation across different axes. We note that there also exist dedicated IL benchmarks [30, 26], but they are aimed at solving challenging robotics tasks rather than evaluating generalisation directly.

There are many machine learning benchmarks that evaluate generalisation outside of IL. For instance, there are several challenging benchmarks for generalisation [31, 12, 13] and meta- or multi-task learning [40] in RL. Unlike MAGICAL, these RL benchmarks have no ambiguity about what the goal is in the training environment, since it is directly specified via a reward function. Rather, the challenge is to ensure that the learnt policy (for model-free methods) can achieve that clearly-specified goal in different contexts (RL generalisation), or solve multiple tasks simultaneously (multi-task RL), or be adapted to new tasks with few rollouts (meta-RL). There are also several instruction-following benchmarks for evaluating generalisation in natural language understanding [29, 34]. Although these are not IL benchmarks, they are similar to MAGICAL in that they include train/test splits that systematically evaluate different aspects of generalisation. Finally, the Abstract Reasoning Corpus (ARC) is a benchmark that evaluates the ability of supervised learning algorithms to extrapolate geometric patterns in a human-like way [11]. Although there is no sequential decision-making aspect to ARC, Chollet [11] claims that solving the corpus may still require priors for "objectness", goal-directedness, and various geometric concepts, which means that methods suitable for solving MAGICAL may also be useful on ARC, and vice versa.

Although we covered some simple methods of improving IL robustness in Section 3, there also exist more sophisticated methods tailored to different IL settings. Meta-IL [15, 25] and meta-IRL [38, 39]

algorithms assume that a large body of demonstrations is available for some set of "train tasks", but only a few demonstrations are available for "test tasks" that might be encountered in the future. Each test task is assumed to have a distinct objective, but one that shares similarities with the train tasks, making it possible to transfer knowledge between the two. These methods are likely useful for multi-task learning in the context of MAGICAL, too. However, it's worth noting that past meta-IL work generally assumes that meta-train and meta-test settings are similar, whereas this work is concerned with how to generalise the intent behind a few demonstrations given in one setting (the demo variant) to other, potentially very different settings (the test variants). Similar comments apply to existing work on multi-task IL and IRL [18, 10, 14, 3].

## 6    Conclusion

In this paper, we introduced the MAGICAL benchmark suite, which is the first imitation learning benchmark capable of evaluating generalisation across distinct, semantically-meaningful axes of variation in the environment. Unsurprisingly, results for the MAGICAL suite confirm that single-task methods fail to transfer to changes in the colour, shape, position and number of objects. However, we also showed that image augmentations and perspective shifts only slightly ameliorate this problem, and multi-task training can sometimes make it *worse*. This lack of generalisation stands in marked contrast to human imitation: even 14-month-old infants have been observed to generalise demonstrations of object manipulation tasks across changes in object colour and shape, or in the appearance of the surrounding room [4]. Closing the gap between current IL capabilities and human-like few-shot imitation could require significant innovations in multi-task learning, action and state representations, or models of human cognition. The MAGICAL suite provides a way of evaluating such algorithms which not only tests whether they generalise well "on average", but also shines a light on the specific kinds of generalisation which they enable.

## 7    Broader impact

This paper presents a new benchmark for robust IL and argues for an increased focus on algorithms that can generalise demonstrator intent across different settings. We foresee several possible follow-on effects from improved IL robustness:

**Economic effects of automation**  Better IL generalisation could allow for increased automation in some sectors of the economy. This has the positive flow-on effect of increased economic productivity, but could lead to socially disruptive job loss. Because our benchmark focuses on robust IL in robotics-like environments, it's likely that any effect on employment would be concentrated in sectors involving activities that are expensive to record. This could include tasks like surgery (where few demonstrators are qualified to perform the task, and privacy considerations make it difficult to collect data) or packaging retail goods for postage (where few-shot learning might be important when there are many different types of goods to handle).

**Identity theft and model extraction**  More robust IL could enable better imitation of *specific* people, and not just imitation of people in general. This could lead to identify theft, for instance by mimicking somebody's speech or writing, or by fooling biometric systems. Because this benchmark focuses on control and manipulation rather than media synthesis, it's unlikely that algorithms designed to solve our benchmark will be immediately useful for this purpose. On the other hand, this concern is still relevant when applied to machine behaviour, rather than human behaviour. In NLP, it's known that weights for ML models can be "stolen" by observing the model's outputs for certain carefully chosen inputs [28]. Similarly, more robust IL could make it possible to clone a robot's policy by observing its behaviour, which could make it harder to sell robot control algorithms as standalone products.

**Learnt objectives**  Hadfield-Menell et al. [21] argues that it is desirable for AI systems to infer their objectives from human behaviour, rather than taking them as fixed. This can avoid problems that arise when an agent (human, robot, or organisation) doggedly pursues an easy-to-measure but incorrect objective, such as a corporate executive optimising for quarterly profit (which is easy to measure) over long-term profitability (which is actually desired by shareholders). IL makes it possible to learn objectives from observed human behaviour, and

more robust IL may therefore lead to AI systems that better serve their designers' goals. However, it's worth noting that unlike, say, HAMDPs [16] or CIRL games [21], IL cannot request clarification from a demonstrator if the supplied demonstrations are ambiguous, which limits its ability to learn the right objective in general. Nevertheless, we hope that insights from improved IL algorithms will still be applicable to such interactive systems.

## Acknowledgments and Disclosure of Funding

We would like to thank reviewers for helping to improve the presentation of the paper (in particular, clarifying the distinction between traditional IL and robust IL), and for suggesting additional related work and baselines. This work was supported by a Berkeley Fellowship and a grant from the Open Philanthropy Project.

## Footnotes

[1]Multi-task imitation learning algorithms: `https://github.com/qxcv/mtil/`

[2]Benchmark suite and links to data: `https://github.com/qxcv/magical/`

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
