[Supplementary Material]

# A  Additional benchmark details

In this section we provide more details about our benchmark tasks, including horizons, scoring functions, and so on. We also list the test variants available for each task in Table 2.

| Task | Test variant | | | | | | |
|---|---|---|---|---|---|---|---|
| | Jitter | Layout | Colour | Shape | CountPlus | Dynamics | All |
| MoveToCorner | ✓ | ✗ | ✓ | ✓ | ✗ | ✓ | ✓ |
| MoveToRegion | ✓ | ✓ | ✓ | ✗ | ✗ | ✓ | ✓ |
| MatchRegions | ✓ | ✓ | ✓ | ✓ | ✓ | ✓ | ✓ |
| MakeLine | ✓ | ✓ | ✓ | ✓ | ✓ | ✓ | ✓ |
| FindDupe | ✓ | ✓ | ✓ | ✓ | ✓ | ✓ | ✓ |
| FixColour | ✓ | ✓ | ✓ | ✓ | ✓ | ✓ | ✓ |
| ClusterColour | ✓ | ✓ | ✓ | ✓ | ✓ | ✓ | ✓ |
| ClusterType | ✓ | ✓ | ✓ | ✓ | ✓ | ✓ | ✓ |

Table 2: Available variants for each task. Some variants are not defined for certain tasks because they may make task completion impossible, make task completion trivial (i.e. the null policy often completes the task), or do not provide a meaningful axis of variation (e.g. MoveToRegion does not feature any blocks, and so there are no shapes to randomise).

## A.1  Action and observation space

(a) Egocentric                                  (b) Allocentric

Figure 3: Egocentric and allocentric views of a demonstration on MoveToRegion. The four $96{\times}96$ RGB frames shown in each subfigure would normally be stacked together along the channels axis before being passed to an agent policy or discriminator.

We use the same discrete action space for all tasks. Although this benchmark was inspired by robotic IL, where the underlying action space is generally continuous, we opted to use discrete actions so that we could elicit human demonstrations using only a standard keyboard. The underlying state space is still continuous, so each discrete action applies a preset combination of forces to the robot, such as a force that pushes the gripper arms together, or a force that moves the robot forward or backward. In total, the agent has 18 distinct actions. These are formed from the Cartesian product of two gripper actions (push closed/allow to open), three longitudinal motion actions (forward/back/stop), and three angular motions (left/straight/right).

We use the same image-based observation space for each task. In all of our experiments, we provide the agent with stacked $96{\times}96$ pixel RGB frames depicting the workspace at the current time step and three preceding time steps. At our 8Hz control rate, this corresponds to around 0.5s of interaction context. Using an image-based observation space makes it easy to generalise policies and discriminators across different numbers and types of objects, without having to resort to, e.g., graph networks or structured learning. An image-based observation space also means that the agent gets access to a similar representation as the human demonstrator. This makes it possible to resolve ambiguities and improve generalisation by exploiting features of the human visual system, as we do when we apply the small image augmentations described in Appendix B.

By default, observations employ an egocentric (robot-centred) perspective on the workspace, as illustrated in Fig. 3a. Unlike the allocentric perspective, depicted in Fig. 3b, the egocentric often does not allow the agent to observe the full workspace. However, we found that an egocentric perspective resulted in faster training and better generalisation, as we note in the ablations of Section 4. Similar benefits to generalisation were previously observed by Hill et al. [22].

## A.2 Detailed task descriptions

### A.2.1 MoveToCorner (MTC)

Figure 4: A demonstration on MoveToCorner.

In MoveToCorner, the robot must push a single block from the bottom right corner of the workspace to the top left corner of the workspace. Test variants are also constrained so that there is only ever one block, and it always starts close to the bottom right corner of the workspace. These constraints preclude use of the CountPlus test variant, since block count cannot be changed without making the task ambiguous. It also precludes use of the Layout variant, since fully randomising block position might make the desired block location ambiguous (e.g. pushing the block into top left corner versus pushing it to the opposite side of the workspace). The horizon for all variants is $H = 80$ time steps.

Trajectories receive a score of $S(\tau) = 1$ if the block spends the last frame of the rollout within $\sqrt{2}/2$ units of the top left corner of the workspace (the whole workspace is 2×2 units). $S(\tau)$ decays linearly from 1 to 0 as the block moves from inside that region to more than $\sqrt{2}$ units away from the corner.

### A.2.2 MoveToRegion (MTR)

Figure 5: A demonstration on MoveToRegion.

The objective of the MoveToRegion task is for the robot to drive inside a goal region placed in the workspace. There is only ever one goal region, and no blocks are present in the train or test variants. Hence the CountPlus and Shape variants are not applicable. However, the Colour variant is still applicable. as it randomises the colour of the goal region. The horizon is set to $H = 40$.

Scoring for MoveToRegion is binary. If at the end of the episode, the centre of the robot's body is inside the goal region, then it receives a score of 1. Otherwise it receives a score of 0.

### A.2.3 MatchRegions (MR)

Figure 6: A demonstration on MatchRegions.

In MatchRegions, the agent is confronted with a single goal region and several blocks of different colours. The objective is to move all (and only) blocks of the same colour as the goal region into the goal region. All test variants are applicable to this version, although CountPlus only randomises the number of blocks (and not the number of goal regions) in order to avoid ambiguity about which goal region(s) the robot should fill with blocks. The horizon is fixed to $H = 120$.

At the end of a trajectory $\tau$, the robot receives a score of

$$S(\tau) = \underbrace{\frac{|\mathcal{T} \cap \mathcal{R}|}{|\mathcal{T}|}}_{\text{Target bonus}} \times \underbrace{\left(1 - \frac{|\mathcal{D} \cap \mathcal{R}|}{|\mathcal{R}|}\right)}_{\text{Distractor penalty}}.$$

Here $\mathcal{T}$ is the set of *target blocks* of the same colour as the goal region, $\mathcal{D}$ is the set of *distractor blocks* of a different colour, and $\mathcal{R}$ is the set of blocks inside the goal region in the last state $s_T$ of the rollout $\tau$. The agent gets a perfect score of 1 for placing all the target blocks and none of the distractors in the goal region. Its score decreases for each target block it fails to move to the goal region (target bonus) and each distractor block it improperly places in the goal region (distractor penalty).

### A.2.4  MakeLine (ML)

Figure 7: A demonstration on MakeLine.

The objective of the MakeLine task is to arrange all of the blocks in the workspace into a line. The orientation and location of the line are ignored, as are the shapes and colours of the blocks involved. The horizon for this task is $H = 180$.

Scoring for MakeLine is a function of the relative positions of blocks in the final state of a trajectory, and in particular the number of blocks that form the largest identifiable "line". To identify lines of blocks, we use a line-fitting methods that is similar in spirit to RANSAC [7], but with constraints to ensure that blocks are spread out along the length of the line rather than "bunching up". Our definition of what constitutes a line is based on a relation between triples of blocks: we say that a block $b_k$ is considered to be part of a line between blocks $b_i$ and $b_j$ if:

1. $b_k$ **is an inlier:** it must lie a distance of at most $d_i = 0.18$ units from the (geometric) line that links $b_i$ and $b_j$ (recall that the workspace is $2 \times 2$ units).

2. $b_k$ **is close to other blocks in the line:** if $b_k$ is not the first or last block in the line of blocks, then it must be a distance of at most $d_c = 0.42$ units from the previous and next blocks. Here the distance is measured along the direction of the geometric line between $b_i$ and $b_j$. That is, by projecting the previous and next inliers onto the geometric line between $b_i$ and $b_j$, then taking the distance between those projections and the projection for $b_k$.

Note that if $b_i$ and $b_j$ are a long way apart, then there may be several subsets of inliers for the line between $b_i$ and $b_j$, each of which is separated from the other subsets than $d_c$ units. For any given pair of blocks $(b_i, b_j)$, let $\#(b_i, b_j)$ be the number of blocks that form the *largest* such subset for the line between $b_i$ and $b_j$ (potentially including $b_i$ and/or $b_j$, if they are close enough to the other inliers). Further, let $n$ be the number of blocks in the workspace, and $m = \max_{i,j} \#(b_i, b_j)$ be the largest number of blocks on a line between any two blocks in the final state. If $m = n$, then all blocks belong to the same line, and so $S(\tau) = 1$. If $m = n - 1$, then exactly one block is not a part of the largest identifiable line, and $S(\tau) = 0.5$. Otherwise, if $m < n - 1$, the agent receives a score of $S(\tau) = 0$.

### A.2.5 FindDupe (FD)

Figure 8: A demonstration on FindDupe.

FindDupe presents the agent with a goal region that has a single "query" block inside it, along with a mixture of blocks outside the goal region. The agent's objective is to locate at least one block outside the goal region with the same shape and colour as the query block, and push it inside the goal region. Variants are constrained so that there is only ever one goal region and query block, and so that there is at least one duplicate of the query block outside the goal region. The horizon for this task is $H = 100$.

The score for this task is a function of the set of blocks present in the goal area at the end of the trajectory. Let $\mathcal{R}$ denote the set of blocks inside the region at the end of the episode, let $\mathcal{T}$ denote the set of all *target* blocks with the same shape and colour as the query block, and let $\mathcal{D}$ denote the set of all *distractor* blocks with a different shape or colour. Further, let $q$ refer to the original query block. The score $S(\tau)$ for a trajectory is

$$S(\tau) = \underbrace{\mathbb{I}[q \in \mathcal{R}] \times \mathbb{I}[\mathcal{T} \cap \mathcal{R} \neq \varnothing]}_{\text{Query satisfied?}} \times \underbrace{\left(1 - \frac{|\mathcal{D} \cap \mathcal{R}|}{|\mathcal{R}|}\right)}_{\text{Distractor penalty}} .$$

The first factor ensures that the query block remains inside the goal region. The second factor ensures that at least one other block with the same attributes as the query block is in the goal region. Finally, the last factor creates a penalty for pushing distractor blocks into the goal region.

### A.2.6 FixColour (FC)

Figure 9: A demonstration on FixColour.

FixColour variants always include several non-overlapping goal regions, each containing a single block. Exactly one of those blocks will be of a different colour to its enclosing goal region; we'll call this the "mismatched block". The agent's objective is to identify the mismatched block and push it out of its goal region, into an unoccupied part of the workspace, thereby "fixing" the mismatch. The horizon for this task is $H = 60$.

Scoring for FixColour is binary. A score of $S(\tau) = 1$ is given if, in the final state, the mismatched block is not in its original goal region. All other goal regions must contain exactly the same block that they started with (and in particular cannot contain the mismatched block). If any of these conditions is not satisfied, then the score is zero.

### A.2.7 ClusterColour (CC) and ClusterShape (CS)

Figure 10: Demonstrations on ClusterColour (top) and ClusterShape (bottom).

In both ClusterColour and ClusterShape, the workspace is initially filled with a jumble of blocks of different colours and types, and the agent must push the blocks into clusters according to some attribute. For ClusterColour, blocks should belong to the same cluster iff they have the same colour, while ClusterShape applies the analogous criterion to block shape. All variants are applicable to these tasks. Because these tasks require interaction with most or all blocks in the workspace, the horizon is set to $H = 320$ (40s at 8Hz).

The score $S(\tau)$ takes the same form for both ClusterColour and ClusterShape, but with a different attribute-of-interest (either colour or shape). Specifically, $S(\tau)$ is computed by applying a K-means-like objective to the final state $s_T$ of the rollout $\tau$. For each value $a$ of the attribute-of-interest (either red/green/blue/yellow for ClusterColour or square/circle/pentagon/star for ClusterShape), a *centroid* $x_a$ is computed from the mean positions of blocks with the corresponding attribute value. Formally, this is

$$x_a = \frac{1}{|\mathcal{B}_a|} \sum_{b \in \mathcal{B}_a} b.\text{pos} \, ,$$

where $\mathcal{B}_a$ is the set of blocks with the relevant attribute set to value $a$, and $b.\text{pos}$ is the position of block $b$ in state $s_T$. In order for an individual block $b$ with relevant attribute value $a$ to be considered correctly clustered, the squared distance

$$d(b, a) = \|b.\text{pos} - x_a\|_2^2$$

between it and its associated centroid must be at most a third the squared distance $d(b, a')$ between it and the nearest centroid for any other attribute value $a'$. Specifically, we must have

$$d(b, a) < \frac{1}{3} \min_{a' \neq a} d(b, a') \, .$$

When 50% or fewer of blocks are correctly clustered in the final state of a trajectory, the score $S(\tau) = 0$. As the fraction of correctly clustered blocks increases from 50% up to 100%, the score $S(\tau)$ increases linearly from 0 to 1.

## B  Addition experiment details

This section documents the full set of hyperparameters we used for BC and GAIL, along with additional details on how we collected and preprocessed our demonstrations.

**Dataset and data preprocessing details**   We collected training datasets of 25 demonstration trajectories for the demonstration variant of each task. These trajectories were recorded by the authors to show several distinct strategies for solving the task within the demonstration variant. For instance,

in ClusterColour, there are demonstrations that place clusters in different locations or construct them in a different order. Appendix A shows a single demonstration for each task in the dataset.

Each algorithm run used only 10 of the 25 total trajectories for each task (or 10 trajectories for each task, in the multi-task case). The subset of 10 trajectories was sampled at random based on the seed for that run. We did not hold out any trajectories for testing or validation; rather, our evaluation is based on the test variant scores assigned to the trained policy produced by each algorithm. For all policies, value functions, and discriminators, we constructed an observation by concatenating four temporally adjacent RGB frames along the channels axis, scaling the pixel values into the $[0, 1]$ range, and resizing the stacked frames to 96×96 pixels. For BC, we performed the additional preprocessing step of removing samples with noop actions from the demonstration dataset, as described below.

**Evaluation details**    For single-task BC and GAIL, we do five training runs on each task with different random seeds. After each run, we take the trained policy, use it to perform 100 rollouts on each test variant of the original task, and retain the mean scores from those 100 trajectories. In tables, we report "mean score ± standard deviation of score", where the mean and standard deviation are taken over the mean evaluation scores for each of the five runs on each algorithm and task. Multitask evaluations are similar, except we pool data from all tasks together, and consequently only perform five runs in total rather than five runs per task. To reduce variance, we used the same five random seeds (and consequently the same five subsets of 10 training trajectories each) for all algorithms and tasks.

**Default augmentation set**    Throughout the text, we refer to noise, translation, rotation, and colour jitter augmentations. Concretely, these augmentations involved the following operations:

- **Noise:** Each (RGB) channel of each pixel is independently perturbed by additive noise sampled from $\mathcal{N}(0, 0.01)$.

- **Translation:** The image is mirror-padded and randomly translated along the $x$ and $y$ axes by up to 5% of their respective range (so ±4.8px, for 96×96 pixels).

- **Rotation:** Image is mirror padded and then rotated around its centre by up to ±5 degrees.

- **Colour jitter:** For this augmentation, images are translated to the CIELab colour space. The luminance channel is rescaled by a randomly sampled factor between 0.99 and 1.01, while the $a$ and $b$ channels are treated as a 2D vectors and randomly rotated by up to ±0.15 radians. We use the same luminance scaling factor and colour rotation for each pixel an a given image. After these operations, images are converted back to RGB.

For the translation, rotation, and colour jitter augmentations, we apply the same randomly sampled transformation to each image in a four-image "stack" of frames, but different, independently sampled transformations to each stack in a training batch.

**Single- and multi-task BC hyperparameters**    The hyperparameters for BC are given in Table 3. BC hyperparameters were manually tuned to ensure that losses plateaued on most single-task problems. Note that hyperparameters for single- and multi-task learning were identical. In particular, we retained the same batch size for multi-task experiments, and randomly sampled demonstration states from each task with a weighting that ensured equal representation from all tasks. Initially, we found that training BC to convergence would cause the policy to get "stuck" in states where the most probable demonstrator action was a noop action. We avoided this problem by removing all state/action pairs with noop actions from the dataset in our BC experiments; we did not do this in our GAIL experiments.

**Single- and multi-task GAIL hyperparameters**    Hyperparameters for GAIL are listed in Table 4. For policy optimisation, we used the PPO implementation from rlpyt [36]; PPO hyperparameters that are not listed in Table 4 took their default values in rlpyt. To prevent value and advantage magnitudes from exploding in PPO, we normalised rewards produced by the discriminator to have zero mean and a standard deviation of 0.1, both enforced using a running average and variance updated over the course of training. Again, multi-task hyperparameters were the same as single-task hyperparameters, and we split each policy and discriminator training batch evenly between the tasks.

| Hyperparameter | Value | Range Considered |
|---|---|---|
| Total opt. batches | 20,000 | 5,000–20,000 |
| Batch size | 32 | - |
| SGD learning rate | $10^{-3}$ | - |
| SGD momentum | 0.1 | - |
| Policy augmentations | Noise, trans., rot., colour jit. | - |

Table 3: Hyperarameters for BC experiments.

| Hyperparameter | Value | Range Considered |
|---|---|---|
| Policy (PPO) | | |
|     Sampler batch size | 32 | 16 to 64 |
|     Sampler time steps | 8 | 8 to 20 |
|     Opt. epochs per update | 12 | 2 to 10 |
|     Opt. minibatch size | 64 | 42 to 64 |
|     Initial Adam step size | $6 \times 10^{-5}$ | $10^{-6}$ to $10^{-3}$ |
|     Final Adam step size | 0 (lin. anneal) | - |
|     Discount $\gamma$ | 0.8 | 0.8 to 1.0 |
|     GAE $\lambda$ | 0.8 | 0.8 to 1.0 |
|     Entropy bonus | $10^{-5}$ | $10^{-6}$ to $10^{-4}$ |
|     Advantage clip $\epsilon$ | 0.01 | 0.01 to 0.2 |
|     Grad. clip $\ell_2$ norm | 1.0 | - |
|     Augmentations | N/A | - |
| Discriminator | | |
|     Batch size | 24 | - |
|     Adam step size | $2.5 \times 10^{-5}$ | $10^{-5}$ to $5 \times 10^{-4}$ |
|     Augmentations | Noise, trans., rot., colour jit. | - |
|     $\lambda_{\text{w-gp}}$ (WGAIL-GP) | 100 | - |
| Misc. | | |
|     Disc. steps per PPO update | 12 | 8 to 32 |
|     Total env. steps of training | $10^6$ | $5 \times 10^5$ to $5 \times 10^6$ |
|     Reward norm. std. dev. | 0.1 | - |

Table 4: Hyperarameters for GAIL experiments.

**Apprenticeship learning baseline** In addition to our BC and (W)GAIL baselines, we also attempted to train a feature expectation matching Apprenticeship Learning (AL) baseline [1, 24]. Given a feature function $\Phi : \mathcal{S} \times \mathcal{A} \to \mathbb{R}^n$, the goal of AL is to find a policy $\pi_\theta$ that matches the expected value of the feature function $\Phi$ under the demonstration distribution with its expected value under the novice distribution. That is, we seek a $\pi_\theta$ such that $\mathbb{E}_{\pi_\theta} \Phi(s, a) = \mathbb{E}_{\mathcal{D}} \Phi(s, a)$. Matching feature expectations is equivalent to finding a policy $\pi_\theta$ that drives the cost

$$\sup_{\|w\| \leq 2} \left[ \mathbb{E}_{\mathcal{D}} w^T \Phi(s, a) - \mathbb{E}_{\pi_\theta} w^T \Phi(s, a) \right] \tag{1}$$

to zero. Observe that if $w^*$ is a weight vector that attains the supremum in Eq. (1), then

$$-\nabla_\theta \mathbb{E}_{\pi_\theta} w^{*T} \Phi(s, a)$$

is a subgradient of Eq. (1) with respect to the policy parameters $\theta$. Thus, our training procedure consisted of alternating between optimising Eq. (1) to convergence with respect to $w$, and taking a PPO step on the policy parameters using the reward function $r(s, a) = w^{*T} \Phi(s, a)$ (recall that RL maximises return, but we want to minimise Eq. (1)). To optimise $w$, we used 512 samples from the expert and the novice, and to optimise $\pi_\theta$, we used the same generator hyperparameters as our GAIL runs. This single-task AL baseline is denoted "AL (ST)" in results tables.

Figure 11: Base architecture for policies, value functions and discriminators. "$n$c" is used as an abbreviation for "$n$ channels". Refer to main text for a discussion of which networks use the optional features (batch norm, action input), and for a description of the final layer for each network type.

The feature function $\Phi$ used for AL was acquired by removing the final (logit) layer of our GAIL discriminator network architecture and optimising the remaining layers to minimise an autoencoder loss. In creating the encoder, our only modification to the GAIL discriminator network architecture was to replace the 256-dimensional penultimate layer with a 32-dimensional one, to produce a 32-dimensional feature function $\Phi$. This optimisation was performed for 8,192 size-24 batches of expert data, which we empirically found was enough to get clear reproduction of most input images. After autoencoder pretraining, the encoder weights were kept frozen for the remainder of each training run.

Unfortunately, we could not get AL to produce adequate policies for any task except MoveToCorner. We suspect that the poor performance of AL was due to inadequate autoencoder features. The autoencoder was only trained on expert samples, and we found that for some problems it would not correctly reproduce images of states that were far from the support of the demonstrations. It may be possible to improve results by training the autoencoder on both random rollouts and expert samples, or by training it on more diverse multi-task data.

**Network architecture** Fig. 11 shows the base architecture for all neural networks used in the experiments (including discriminators, policies, and value functions). Some experiments use slight variations on this basic policy architecture for some of the networks:

- The one-hot action input is only used for discriminators, which concatenate the one-hot action representation to the activations of the final convolution layer before performing a forward pass through the linear layers.

- Batch norm is only used for the BC policy and GAIL discriminator, not for the GAIL policy and value function.

- In GAIL experiments, which train a policy via RL, the policy and value function share all layers *except* the final fully-connected layer.

- In multitask experiments, the policy, value function, and discriminator share weights between tasks for all layers except the last. The final layer uses a single, separate set of weights corresponding to each task.

**Computing infrastructure and experiment running time** Experiments were performed on machines with 2× Xeon Gold 6130 CPUs (16 cores each, 2.1GHz base clock), 128–256GB RAM, and 4× GTX 1080-Ti GPUs. Each "run"—that is, the training and evaluation of a specific algorithm on a specific task with one seed—took an average of 10h03m (GAIL) and 32m (BC). It should be noted that these wall time figures were recorded while performing up to 16 runs in parallel on each machine. Because we did not use task-specific training durations, there was little variance in execution time between the different configurations (multi-task, egocentric, allocentric, etc.) of each of the two main base algorithms (BC and GAIL).

# C   Full experiment results

Full results for all methods, along with corresponding ablations, are shown in Table 5, Table 6, Table 7 and Table 8. We abbreviate behavioural cloning as "BC" and generative adversarial IL as "GAIL", while apprenticeship learning is "AL". Single-task methods are denoted with "(ST)" and multi-task methods with "(MT)". "Allo." is for experiments using an allocentric view; all other expeirments use an egocentric view. For GAIL, "WGAIL-GP" denotes a version of GAIL that approximately minimises Wasserstein divergence while using a gradient penalty to encourage 1-Lipschitzness of the discriminator. For augmentation ablations, we use "no trans./rot. aug." to denote removal of translation/rotation; and "no aug." to denote removal of all three default augmentations (colour, translation/rotation, Gaussian noise).

| | **MoveToCorner** | | | | | | | |
| | Demo | Jitter | Layout | Colour | Shape | CountPlus | Dynamics | All |
|---|---|---|---|---|---|---|---|---|
| AL (ST) | 0.00±0.00 | 0.00±0.00 | - | 0.00±0.00 | 0.00±0.00 | - | 0.00±0.00 | 0.00±0.00 |
| BC (MT) | 0.97±0.04 | 0.91±0.02 | - | 0.73±0.16 | 0.98±0.01 | - | 0.92±0.04 | 0.62±0.11 |
| BC (ST) | 0.98±0.04 | 0.86±0.07 | - | 0.96±0.05 | 0.97±0.03 | - | 0.91±0.05 | 0.84±0.06 |
| BC (ST, allo.) | 0.94±0.05 | 0.89±0.04 | - | 0.93±0.04 | 0.97±0.02 | - | 0.90±0.04 | 0.91±0.02 |
| BC (ST, no aug.) | 0.96±0.04 | 0.77±0.09 | - | 0.80±0.06 | 0.81±0.12 | - | 0.86±0.05 | 0.60±0.05 |
| BC (ST, no trans./rot. aug.) | 0.96±0.04 | 0.85±0.04 | - | 0.83±0.14 | 0.88±0.06 | - | 0.94±0.05 | 0.67±0.11 |
| GAIL (MT) | 0.31±0.31 | 0.33±0.31 | - | 0.16±0.09 | 0.34±0.30 | - | 0.30±0.27 | 0.16±0.10 |
| GAIL (ST) | 0.99±0.01 | 0.91±0.06 | - | 0.78±0.10 | 0.95±0.03 | - | 0.95±0.05 | 0.65±0.16 |
| GAIL (ST, allo.) | 1.00±0.00 | 0.82±0.05 | - | 0.90±0.08 | 0.99±0.01 | - | 0.99±0.01 | 0.59±0.12 |
| GAIL (ST, no aug.) | 0.56±0.36 | 0.36±0.23 | - | 0.39±0.30 | 0.34±0.24 | - | 0.46±0.27 | 0.11±0.10 |
| WGAIL-GP (ST) | 0.35±0.24 | 0.22±0.12 | - | 0.17±0.20 | 0.32±0.21 | - | 0.30±0.21 | 0.04±0.05 |

| | **MoveToRegion** | | | | | | | |
| | Demo | Jitter | Layout | Colour | Shape | CountPlus | Dynamics | All |
|---|---|---|---|---|---|---|---|---|
| AL (ST) | 0.51±0.42 | 0.47±0.39 | 0.22±0.17 | 0.21±0.22 | - | - | 0.51±0.41 | 0.09±0.05 |
| BC (MT) | 0.79±0.22 | 0.77±0.26 | 0.52±0.13 | 0.60±0.16 | - | - | 0.81±0.24 | 0.26±0.04 |
| BC (ST) | 0.89±0.11 | 0.88±0.11 | 0.48±0.12 | 0.60±0.13 | - | - | 0.88±0.10 | 0.28±0.06 |
| BC (ST, allo.) | 0.63±0.08 | 0.57±0.14 | 0.09±0.02 | 0.56±0.19 | - | - | 0.61±0.12 | 0.10±0.02 |
| BC (ST, no aug.) | 0.88±0.12 | 0.83±0.11 | 0.44±0.08 | 0.75±0.17 | - | - | 0.87±0.12 | 0.36±0.07 |
| BC (ST, no trans./rot. aug.) | 0.91±0.05 | 0.85±0.10 | 0.44±0.10 | 0.73±0.13 | - | - | 0.89±0.06 | 0.33±0.05 |
| GAIL (MT) | 1.00±0.00 | 0.99±0.02 | 0.69±0.20 | 0.34±0.09 | - | - | 1.00±0.00 | 0.31±0.07 |
| GAIL (ST) | 1.00±0.00 | 1.00±0.00 | 0.71±0.09 | 0.40±0.07 | - | - | 1.00±0.00 | 0.29±0.07 |
| GAIL (ST, allo.) | 1.00±0.00 | 0.98±0.02 | 0.08±0.02 | 0.95±0.02 | - | - | 1.00±0.00 | 0.10±0.03 |
| GAIL (ST, no aug.) | 0.99±0.03 | 0.79±0.08 | 0.34±0.10 | 0.56±0.12 | - | - | 0.93±0.06 | 0.20±0.06 |
| WGAIL-GP (ST) | 0.94±0.03 | 0.87±0.04 | 0.60±0.06 | 0.23±0.01 | - | - | 0.94±0.03 | 0.18±0.04 |

Table 5: Scores for all compared methods on two tasks, reported as "mean (std.)" over five training runs (individual run means were computed with 100 rollouts each). A colour scale (          ) grades mean scores from poor (lightest) to perfect (darkest). See main text in Appendix C for abbreviations.

| | **MatchRegions** | | | | | | | |
| --- | --- | --- | --- | --- | --- | --- | --- | --- |
| | Demo | Jitter | Layout | Colour | Shape | CountPlus | Dynamics | All |
| AL (ST) | 0.00±0.00 | 0.00±0.00 | 0.01±0.01 | 0.00±0.00 | 0.00±0.00 | 0.01±0.01 | 0.00±0.00 | 0.01±0.01 |
| BC (MT) | 0.69±0.10 | 0.60±0.09 | 0.05±0.01 | 0.32±0.06 | 0.65±0.07 | 0.04±0.02 | 0.64±0.09 | 0.04±0.01 |
| BC (ST) | 0.77±0.09 | 0.69±0.11 | 0.07±0.02 | 0.42±0.04 | 0.69±0.10 | 0.07±0.03 | 0.68±0.08 | 0.05±0.02 |
| BC (ST, allo.) | 0.72±0.06 | 0.58±0.14 | 0.01±0.01 | 0.58±0.11 | 0.60±0.09 | 0.02±0.01 | 0.56±0.10 | 0.03±0.02 |
| BC (ST, no aug.) | 0.71±0.05 | 0.49±0.07 | 0.04±0.01 | 0.35±0.05 | 0.59±0.05 | 0.05±0.02 | 0.54±0.05 | 0.04±0.02 |
| BC (ST, no trans./rot. aug.) | 0.75±0.05 | 0.54±0.05 | 0.06±0.01 | 0.34±0.03 | 0.62±0.07 | 0.06±0.01 | 0.63±0.02 | 0.05±0.02 |
| GAIL (MT) | 0.19±0.12 | 0.20±0.10 | 0.05±0.02 | 0.07±0.03 | 0.19±0.12 | 0.02±0.01 | 0.20±0.11 | 0.04±0.01 |
| GAIL (ST) | 0.94±0.03 | 0.92±0.03 | 0.21±0.02 | 0.31±0.10 | 0.93±0.05 | 0.14±0.04 | 0.92±0.04 | 0.14±0.04 |
| GAIL (ST, allo.) | 0.64±0.13 | 0.58±0.11 | 0.01±0.01 | 0.36±0.08 | 0.62±0.11 | 0.01±0.01 | 0.57±0.11 | 0.02±0.02 |
| GAIL (ST, no aug.) | 0.44±0.24 | 0.35±0.18 | 0.04±0.03 | 0.18±0.13 | 0.35±0.20 | 0.03±0.02 | 0.36±0.19 | 0.02±0.01 |
| WGAIL-GP (ST) | 0.32±0.05 | 0.30±0.04 | 0.15±0.04 | 0.07±0.00 | 0.32±0.05 | 0.11±0.03 | 0.28±0.03 | 0.08±0.02 |

| | **MakeLine** | | | | | | | |
| --- | --- | --- | --- | --- | --- | --- | --- | --- |
| | Demo | Jitter | Layout | Colour | Shape | CountPlus | Dynamics | All |
| AL (ST) | 0.00±0.00 | 0.00±0.00 | 0.04±0.01 | 0.00±0.00 | 0.00±0.00 | 0.03±0.01 | 0.00±0.00 | 0.03±0.01 |
| BC (MT) | 0.31±0.07 | 0.29±0.02 | 0.18±0.03 | 0.18±0.03 | 0.30±0.05 | 0.16±0.02 | 0.28±0.07 | 0.14±0.02 |
| BC (ST) | 0.48±0.08 | 0.43±0.07 | 0.20±0.04 | 0.32±0.04 | 0.41±0.05 | 0.18±0.05 | 0.42±0.07 | 0.18±0.04 |
| BC (ST, allo.) | 0.25±0.07 | 0.24±0.04 | 0.04±0.01 | 0.11±0.02 | 0.21±0.07 | 0.03±0.01 | 0.19±0.03 | 0.02±0.01 |
| BC (ST, no aug.) | 0.19±0.03 | 0.12±0.06 | 0.11±0.03 | 0.11±0.02 | 0.14±0.03 | 0.09±0.02 | 0.11±0.05 | 0.09±0.02 |
| BC (ST, no trans./rot. aug.) | 0.23±0.05 | 0.17±0.06 | 0.12±0.04 | 0.13±0.02 | 0.15±0.02 | 0.12±0.06 | 0.14±0.04 | 0.11±0.03 |
| GAIL (MT) | 0.02±0.01 | 0.01±0.01 | 0.06±0.02 | 0.02±0.00 | 0.02±0.01 | 0.05±0.02 | 0.02±0.01 | 0.06±0.03 |
| GAIL (ST) | 0.33±0.19 | 0.33±0.21 | 0.20±0.02 | 0.19±0.07 | 0.27±0.16 | 0.17±0.05 | 0.28±0.15 | 0.17±0.04 |
| GAIL (ST, allo.) | 0.01±0.01 | 0.01±0.01 | 0.05±0.02 | 0.01±0.01 | 0.01±0.01 | 0.04±0.01 | 0.01±0.01 | 0.03±0.01 |
| GAIL (ST, no aug.) | 0.05±0.03 | 0.03±0.02 | 0.05±0.02 | 0.02±0.01 | 0.03±0.02 | 0.05±0.02 | 0.04±0.03 | 0.04±0.02 |
| WGAIL-GP (ST) | 0.10±0.04 | 0.08±0.03 | 0.11±0.01 | 0.06±0.02 | 0.07±0.03 | 0.11±0.04 | 0.08±0.02 | 0.08±0.01 |

Table 6: Additional results; refer to Table 5 for details.

| | FindDupe | | | | | | | |
|---|---|---|---|---|---|---|---|---|
| | Demo | Jitter | Layout | Colour | Shape | CountPlus | Dynamics | All |
| AL (ST) | 0.00±0.00 | 0.00±0.00 | 0.00±0.00 | 0.00±0.00 | 0.00±0.00 | 0.00±0.01 | 0.00±0.00 | 0.00±0.00 |
| BC (MT) | 0.89±0.04 | 0.78±0.12 | 0.05±0.03 | 0.31±0.07 | 0.81±0.07 | 0.03±0.01 | 0.81±0.08 | 0.05±0.03 |
| BC (ST) | 0.89±0.03 | 0.76±0.02 | 0.04±0.01 | 0.60±0.04 | 0.80±0.02 | 0.06±0.02 | 0.77±0.04 | 0.05±0.01 |
| BC (ST, allo.) | 0.93±0.02 | 0.79±0.07 | 0.01±0.01 | 0.72±0.11 | 0.75±0.09 | 0.02±0.01 | 0.74±0.08 | 0.02±0.01 |
| BC (ST, no aug.) | 0.94±0.06 | 0.38±0.06 | 0.06±0.03 | 0.36±0.04 | 0.75±0.08 | 0.04±0.02 | 0.78±0.06 | 0.06±0.02 |
| BC (ST, no trans./rot. aug.) | 0.93±0.03 | 0.45±0.10 | 0.09±0.01 | 0.43±0.05 | 0.81±0.07 | 0.05±0.03 | 0.75±0.09 | 0.06±0.03 |
| GAIL (MT) | 0.43±0.24 | 0.41±0.21 | 0.02±0.02 | 0.06±0.04 | 0.38±0.24 | 0.03±0.02 | 0.39±0.24 | 0.02±0.02 |
| GAIL (ST) | 0.98±0.02 | 0.97±0.01 | 0.10±0.02 | 0.23±0.06 | 0.95±0.04 | 0.05±0.02 | 0.96±0.02 | 0.05±0.01 |
| GAIL (ST, allo.) | 0.95±0.03 | 0.84±0.03 | 0.00±0.00 | 0.50±0.09 | 0.87±0.05 | 0.00±0.01 | 0.90±0.04 | 0.01±0.01 |
| GAIL (ST, no aug.) | 0.46±0.28 | 0.36±0.27 | 0.01±0.01 | 0.10±0.07 | 0.34±0.25 | 0.02±0.01 | 0.37±0.27 | 0.02±0.01 |
| WGAIL-GP (ST) | 0.70±0.09 | 0.46±0.11 | 0.02±0.00 | 0.09±0.04 | 0.64±0.04 | 0.02±0.01 | 0.62±0.03 | 0.03±0.02 |

| | FixColour | | | | | | | |
|---|---|---|---|---|---|---|---|---|
| | Demo | Jitter | Layout | Colour | Shape | CountPlus | Dynamics | All |
| AL (ST) | 0.00±0.01 | 0.03±0.04 | 0.02±0.01 | 0.03±0.03 | 0.00±0.01 | 0.01±0.01 | 0.01±0.01 | 0.01±0.02 |
| BC (MT) | 0.76±0.18 | 0.61±0.14 | 0.15±0.03 | 0.23±0.05 | 0.72±0.18 | 0.13±0.02 | 0.70±0.21 | 0.14±0.07 |
| BC (ST) | 0.62±0.14 | 0.45±0.18 | 0.18±0.04 | 0.19±0.03 | 0.63±0.12 | 0.18±0.04 | 0.71±0.13 | 0.13±0.03 |
| BC (ST, allo.) | 0.88±0.07 | 0.47±0.24 | 0.11±0.03 | 0.32±0.03 | 0.86±0.06 | 0.13±0.04 | 0.84±0.04 | 0.14±0.05 |
| BC (ST, no aug.) | 0.55±0.18 | 0.29±0.12 | 0.19±0.03 | 0.21±0.09 | 0.57±0.18 | 0.22±0.03 | 0.52±0.23 | 0.22±0.02 |
| BC (ST, no trans./rot. aug.) | 0.46±0.14 | 0.28±0.07 | 0.17±0.04 | 0.17±0.07 | 0.48±0.16 | 0.23±0.03 | 0.39±0.13 | 0.20±0.04 |
| GAIL (MT) | 0.99±0.02 | 0.65±0.17 | 0.30±0.08 | 0.21±0.06 | 0.99±0.01 | 0.18±0.10 | 0.95±0.07 | 0.16±0.06 |
| GAIL (ST) | 0.99±0.01 | 0.84±0.07 | 0.32±0.07 | 0.25±0.04 | 0.97±0.03 | 0.19±0.02 | 0.96±0.02 | 0.20±0.01 |
| GAIL (ST, allo.) | 0.98±0.00 | 0.66±0.21 | 0.06±0.02 | 0.36±0.01 | 0.99±0.01 | 0.09±0.04 | 0.98±0.01 | 0.09±0.03 |
| GAIL (ST, no aug.) | 0.98±0.02 | 0.64±0.12 | 0.16±0.05 | 0.27±0.03 | 0.87±0.07 | 0.13±0.04 | 0.84±0.07 | 0.14±0.03 |
| WGAIL-GP (ST) | 0.93±0.04 | 0.72±0.12 | 0.08±0.03 | 0.18±0.02 | 0.93±0.04 | 0.02±0.02 | 0.91±0.05 | 0.05±0.02 |

Table 7: Additional results; refer to Table 5 for details.

| | **ClusterColour** | | | | | | | |
|---|---|---|---|---|---|---|---|---|
| | Demo | Jitter | Layout | Colour | Shape | CountPlus | Dynamics | All |
| AL (ST) | 0.00±0.00 | 0.00±0.00 | 0.00±0.00 | 0.01±0.01 | 0.00±0.00 | 0.00±0.00 | 0.00±0.00 | 0.00±0.00 |
| BC (MT) | 0.11±0.05 | 0.10±0.03 | 0.00±0.01 | 0.01±0.01 | 0.10±0.05 | 0.00±0.00 | 0.09±0.06 | 0.01±0.00 |
| BC (ST) | 0.16±0.05 | 0.17±0.04 | 0.01±0.01 | 0.01±0.00 | 0.17±0.04 | 0.01±0.01 | 0.16±0.05 | 0.01±0.00 |
| BC (ST, allo.) | 0.10±0.05 | 0.11±0.02 | 0.00±0.00 | 0.00±0.00 | 0.10±0.04 | 0.00±0.00 | 0.07±0.03 | 0.00±0.00 |
| BC (ST, no aug.) | 0.04±0.02 | 0.02±0.01 | 0.01±0.01 | 0.01±0.01 | 0.02±0.01 | 0.00±0.00 | 0.04±0.01 | 0.00±0.00 |
| BC (ST, no trans./rot. aug.) | 0.05±0.01 | 0.02±0.00 | 0.00±0.00 | 0.01±0.01 | 0.04±0.02 | 0.00±0.00 | 0.04±0.02 | 0.00±0.00 |
| GAIL (MT) | 0.01±0.01 | 0.01±0.01 | 0.01±0.00 | 0.01±0.00 | 0.01±0.01 | 0.00±0.00 | 0.01±0.01 | 0.00±0.00 |
| GAIL (ST) | 0.12±0.03 | 0.11±0.04 | 0.01±0.00 | 0.01±0.00 | 0.08±0.04 | 0.01±0.00 | 0.11±0.03 | 0.01±0.01 |
| GAIL (ST, allo.) | 0.00±0.00 | 0.00±0.00 | 0.00±0.00 | 0.00±0.00 | 0.01±0.01 | 0.00±0.00 | 0.01±0.01 | 0.00±0.00 |
| GAIL (ST, no aug.) | 0.02±0.01 | 0.01±0.01 | 0.00±0.00 | 0.01±0.01 | 0.02±0.01 | 0.01±0.01 | 0.02±0.01 | 0.00±0.00 |
| WGAIL-GP (ST) | 0.00±0.00 | 0.00±0.00 | 0.00±0.00 | 0.01±0.01 | 0.00±0.00 | 0.00±0.00 | 0.00±0.00 | 0.00±0.00 |

| | **ClusterShape** | | | | | | | |
|---|---|---|---|---|---|---|---|---|
| | Demo | Jitter | Layout | Colour | Shape | CountPlus | Dynamics | All |
| AL (ST) | 0.00±0.00 | 0.00±0.00 | 0.00±0.00 | 0.00±0.00 | 0.00±0.00 | 0.00±0.00 | 0.00±0.00 | 0.01±0.00 |
| BC (MT) | 0.23±0.06 | 0.19±0.05 | 0.00±0.00 | 0.02±0.01 | 0.00±0.00 | 0.00±0.00 | 0.19±0.04 | 0.01±0.00 |
| BC (ST) | 0.35±0.11 | 0.26±0.06 | 0.01±0.01 | 0.06±0.02 | 0.01±0.00 | 0.00±0.00 | 0.31±0.04 | 0.01±0.01 |
| BC (ST, allo.) | 0.21±0.02 | 0.23±0.10 | 0.00±0.00 | 0.13±0.04 | 0.00±0.00 | 0.00±0.00 | 0.16±0.05 | 0.01±0.00 |
| BC (ST, no aug.) | 0.12±0.03 | 0.10±0.02 | 0.00±0.01 | 0.02±0.01 | 0.01±0.01 | 0.00±0.01 | 0.09±0.04 | 0.01±0.01 |
| BC (ST, no trans./rot. aug.) | 0.11±0.05 | 0.11±0.02 | 0.00±0.00 | 0.01±0.01 | 0.00±0.00 | 0.00±0.01 | 0.11±0.04 | 0.01±0.00 |
| GAIL (MT) | 0.01±0.01 | 0.01±0.01 | 0.01±0.00 | 0.00±0.00 | 0.00±0.00 | 0.00±0.00 | 0.01±0.02 | 0.01±0.01 |
| GAIL (ST) | 0.44±0.05 | 0.44±0.08 | 0.01±0.02 | 0.02±0.01 | 0.01±0.00 | 0.00±0.00 | 0.39±0.01 | 0.01±0.00 |
| GAIL (ST, allo.) | 0.00±0.00 | 0.00±0.00 | 0.01±0.01 | 0.00±0.00 | 0.00±0.00 | 0.00±0.00 | 0.00±0.00 | 0.01±0.00 |
| GAIL (ST, no aug.) | 0.01±0.02 | 0.02±0.01 | 0.00±0.00 | 0.00±0.00 | 0.00±0.01 | 0.01±0.00 | 0.01±0.01 | 0.01±0.00 |
| WGAIL-GP (ST) | 0.01±0.00 | 0.01±0.00 | 0.01±0.01 | 0.00±0.00 | 0.00±0.00 | 0.00±0.00 | 0.00±0.00 | 0.00±0.00 |

Table 8: Additional results; refer to Table 5 for details.