[Reviews · NeurIPS 2020]

Review 1

Summary and Contributions: The paper introduces a set of benchmark tasks for imitation learning, with the aim of specifically testing out (1) robustness and (2) generalization. ========================================== Update after author response. Thanks for answering my points and the new experiments. I am changing my score to 6. If the paper is accepted, please make sure the new experiments are included in the camera-ready version. The main reason the paper isn't getting a higher score is weakness 2 - the discussion about why the benchmark is necessary and how it is designed to tease out differences between algorithms could be made much more extensive for an even stronger submission.

Strengths: 1. There is a shortage of good benchmarks in imitation learning literature. Most work uses MuJoCo benchmarks, which require a proprietary simulator and do not test all useful properties of IL algorithms. 2. Robustness and generalization in IL are crucial for deploying it widely.

Weaknesses: 1. It is not clear to me whether the proposed benchmarks are evaluating imitation learning (IL) or robust imitation learning (robust IL). The difference is the standard IL assumes that the expert data and is obtained from an MDP with exactly the same dynamics and the test MDP. Robust IL assumes that we will get a perturbed MDP at test time (where the definition of the perturbation changes depending on the meaning of "robust"). Currently, the paper seems to argue that it is testing imitation learning but is actually testing robust imitation learning. This has consequences in the experiments section. Of course GAIL and BC will fail in the robust IL setting because they assume they get the same transition dynamics at test time. I agree with the authors that robustness is essential in practice, but I think the authors overload the term "imitation learning" too much. 2. Typically, when a benchmark is introduced, it is useful to argue that the new benchmark induces a reasonable ordering on algorithms (or some multi-dimensional generalization of an ordering). In this paper, that argument isn't made very well. Empirically, I think it would be useful to test more algorithm variants. I would probably add (apprenticeship learning (with pre-learned features) and at least one other variant of GAIL (with Wasserstein or MMD divergence). This would allow you to make a fuller comparison. 3. Performance of BC ang GAIL on the proposed benchmarks has a huge level of noise (Table 1). This means that evaluating the benchmarks is expensive. This may well be necessary and justified (benchmarks requiring nontrivial generalization will have higher noise in general), but you should argue for this a bit more.

Correctness: I did not identify any correctness issues in the paper.

Clarity: The paper is clearly written.

Relation to Prior Work: I would definitely mention [1] - is is a very well-known recent application of the idea of robustness in RL (although they did not invent the idea of course). Of course they did vanilla RL, not IL, but the two settings are related. The paper shares many ideas (swapping colors etc). [1] Akkaya, I., Andrychowicz, M., Chociej, M., Litwin, M., McGrew, B., Petron, A., ... & Schneider, J. (2019). Solving rubik's cube with a robot hand. arXiv preprint arXiv:1910.07113.

Reproducibility: Yes

Additional Feedback: Overall, I think working on better IL benchmarks is a great idea. The current way the idea was executed has some issues though. If you address the points in the weakness section in the rebuttal, I will update my score. Also, if you disagree with any of my points please don't be shy about it.


Review 2

Summary and Contributions: “One barrier to better generalisation in IL is a lack of appropriate benchmarks.” This appears to be well defended with relevant citations. Contributes a new multi-task IL benchmark: MAGICAL * Separate training and test variants * Clearly distinguish between different types of transfer * Explicitly designed to enable knowledge reuse across tasks

Strengths: The primary strength of this work is that it provides a benchmark which is sorely needed for IL. This makes it highly relevant to the NeurIPS community. The proposed benchmark is also well presented, with precise characterization of test and training conditions, expected types of generalization, and baseline results. Although the benchmark takes place in a relatively simple environment, it still appears to be relevant to measuring generalization performance of state-of-the-art IL methods. The empirical characterization of the baseline results is also sufficient.

Weaknesses: The biggest limitation of the proposed benchmark is that it is only relevant to pure imitation learning methods. It is unclear how, if at all, this benchmark can be applied to methods that need additional information from an expert (such as DAgger). This limitation is understandable given the wide variety of different ways in which additional information might be acquired from an expert (for example, active learning methods, HG-DAgger, Learning from Intervention), which make constructing a fair benchmark for these methods difficult. The other limitation of this work is that some of the presented information is not clearly actionable. In particular, the multi-task results are worse than the single-task results, and are probably not worth discussing in as much detail.

Correctness: The claims appear to be correct and well supported. The empirical methodology also appears to be correct.

Clarity: The paper is very clear in nearly every respect. There are a few minor awkward bits of writing, such as repetition between the introduction and related work, but the overall quality is still high.

Relation to Prior Work: It is clear how this work differs from previous contributions, although there are a few additional related works it could cite.

Reproducibility: Yes

Additional Feedback: 2 MAGICAL: Systematically evaluating generalisation in IL Describes precisely the structure of MAGICAL: m unique tasks Each task has a non-markovian scoring function S (from 0 to 1), a single demonstration variant, and a set of test variants. Each variant corresponds to a single reward-free MDP precisely. Describes the possible environment contents. Describes the eight different tasks in MAGICAL. Describes the seven types of changes used to produce test variants. 3 Data-efficient intent disambiguation Reviews the generalization performance of two common algorithms on the benchmark (BC and GAIL), as well as relatively straightforward multi-task extensions of them. Mention data-augmentations frequently used to improve generalization. Also mentions ego-centric vs allocentric views, since ego-centric views are known to typically improve generalization. 4 Experiments Presents results on the two baseline algorithms with various alterations mentioned in section 3. The results appear to be consistent with what one would expect, with both IL algorithms performing relatively well (but not optimally) on the single-task demonstration variants, and generalizing a somewhat low but non-zero amount to the test variants. Counter-intuitive results, such as negative multi-task transfer, are briefly expounded upon. 5 Related work Refers to several prior works that use their own ad-hoc benchmarks, as well as a handful of RL generalization benchmarks. Should cite Yu et. al. 2019, Meta-World 6 Conclusion Briefly recaps the primary features of the benchmark. 7 Broader impact Describes three possible follow-on effects of improved generalization in IL algorithms in sufficient detail.


Review 3

Summary and Contributions: ===Edit after rebuttal=== I am raising my score to 7 after reading the rebuttal and other reviews. I appreciate the authors for clarifying my misunderstanding about "task solvable by existing IL algorithms": it is referring to *demonstration* variants and not *test* variants, and pointing out that the *test* variants is actually hard to solve. Second, it is nice to know that the code can be easily integrated with Gym, which clarifies my other concern in ease-of-use. Also to mention that the author's revision regarding R1's weakness point 2 is appealing, which has helped me understand more about the use case of MAGICAL. === This paper presents the MAGICAL benchmark suite for the purpose of providing a systematic evaluation of generalization on imitation learning (IL) algorithms. First, the authors identified the lack of benchmarks that considers the generalization among IL algorithms, mostly due to the lack of train/test split in RL environments (usually the demonstration is performed in the same environment as the imitator). Several tasks that disentangled different aspects (e.g., shapes, color, movement, etc.) were then proposed to address this issue. Second, by conducting experiments over two baseline IL methods (BC and GAIL), the authors (unsurprisingly) found that the agent overfit significantly to the demonstration, and standard methods are not sufficient to eliminate overfitting. The need for such a benchmark environment for evaluating IL methods is well motivated.

Strengths: The proposed new benchmark system was described in detail, with many different aspects been considered when designing the tasks. I like the idea of explicitly separate each component of a task into colors/shapes/movements etc so that it provides an easy way for other researchers to customize the environment to their needs. The experiments on the two IL methods, BC and GAIL, established a good baseline for other researchers to conveniently compare with different methods of their choice (much like the OpenAI baselines).

Weaknesses: The overall MAGICAL design process itself is certainly sound. However, I'm concerned that all the tasks are been made very artificial and even if we found a good IL that works well in this particular domain, how applicable it would be to "feeding their pet salamander"? To be specific, line 119-123, the tasks were designed specifically to be suitable for meta- and multi-task learning, and also is guaranteed to be solvable by existing IL algorithms. I actually think a new benchmark like this should do the opposite: the designed tasks should not be solvable by current methods, which can then encourage the community to take up the challenge. One example of such is that the Atari game of Montezuma's Revenge got a lot of attention because it was hard to play, and many good exploration methods have been developed. In addition, it is not difficult to modify the existing benchmarks like OpenAI gym and Deepmind Lab to suit the need of train/test split and for IL generalization. In my opinion, a better approach could be to make the existing frameworks easier to use (e.g., a wrapper over OpenAI gym that enables color change), rather than making a new one.

Correctness: All methods are sound.

Clarity: The paper is well written and easy to read.

Relation to Prior Work: Related work was appropriately surveyed, the similarity and differences to MAGICAL were clearly stated.

Reproducibility: Yes

Additional Feedback: Overall I like the idea of the paper, and a lot of aspects have been taken into account in the design. I would only argue that the community probably does not need more "scattered" resources, but rather should build on existing tools, making them easier to use.


Review 4

Summary and Contributions: Please see reviewer #2.

Strengths: Please see reviewer #2.

Weaknesses: Please see reviewer #2.

Correctness: Please see reviewer #2.

Clarity: Please see reviewer #2.

Relation to Prior Work: Please see reviewer #2.

Reproducibility: Yes

Additional Feedback:

[Author Response · NeurIPS 2020]

Thank you to all reviewers for their thorough feedback. We're pleased that reviewers appear to endorse the overarching idea of the MAGICAL benchmark ("working on better IL benchmarks is a great idea", "sorely needed", "a lot of aspects have been taken into account"). We're also happy that reviewers found the writing clear and our claims correct. Reviewers raised several valid technical concerns and presentation issues, which we will address below.

**Reviewer 1:** It's good to hear R1 believes that "working on better IL benchmarks is a great idea". The Rubik's Cube ADR paper that R1 mentioned (Akkaya et al.) is relevant to MAGICAL, and we will include it in the camera-ready. Further, we hope that the following points will address the three outstanding issues R1 raised with the paper:

**(1)** "It is not clear to me whether [MAGICAL is] evaluating imitation learning (IL) or robust imitation learning (robust IL)." — Our aim was to measure the ability of IL algorithms to generalise far from the observed training data. The term "robust imitation learning" is a good way to avoid terminological confusion, and we will adopt it in the final revision.

**(2)** ". . . it is useful to argue that the new benchmark induces a reasonable ordering on algorithms . . ." — This is a good point. We ran two of R1's suggested algorithms (Wasserstein GAIL-GP, and apprenticeship learning on autoencoder features) on a subset of three tasks (with fewer seeds and time steps), and show the results in Table AR1. We will include full results in the camera-ready. We did not find that either method generalised better than GAIL. This is perhaps to be expected: most existing IL algorithms do not specifically aim to improve generalisation (or robustness), and so it is not clear a priori what would constitute "reasonable ordering" of methods for this generalisation benchmark. We originally focused on multi-task variants, different views, and augmentation ablations in our experiments because we believed those axes would directly affect the ability of the network to generalise far from the training distribution.

**(3)** "Performance . . . on the proposed benchmarks has a huge level of noise (Table 1)." — The standard deviations shown in Table 1 are pooled across all *tasks*, in addition to being pooled across all seeds. Most reported variance is thus due to the different task difficulty levels. Tables 5–8 in the supplement show that variation across seeds for each algorithm is much lower for most individual tasks. We will be careful to emphasise this in the final revision.

**Reviewer 2:** Thank you to R2 for their positive review! We are happy R2 agrees that MAGICAL "provides a benchmark which is sorely needed for IL". Regarding related work, Yu et al.'s Meta-World benchmark is indeed relevant and we will add it to Section 5. We will also trim the overlap between related work and the introduction, and devote less space to multi-task experiments, as requested; this will help free up space for the baselines requested by R1.

**Reviewer 3:** We're glad that R3 likes the idea of the paper and believes that the design process and methods are sound. As we understand it, R3 has two outstanding concerns:

**(1)** First, R3 raised concerns over our remark on lines 119–123 regarding tasks being "solvable by existing IL algorithms". We would like to emphasise that we only designed the *demonstration* variant of each task to be easy to solve. This allows researchers to focus on the fundamental challenge of MAGICAL, which is generalisation to the *test* variants. Tables 5–8 in the supplement show that existing algorithms fail to solve many of our test variants, even when they can solve the demonstration variant (e.g. the Layout variant of MatchRegions, ClusterColour, etc.). This presents a clear challenge to the community, which we believe is a prerequisite step to more ambitious tasks.

**(2)** Second, R3 suggested that "a better approach could be to make the existing frameworks easier to use . . . rather than making a new one." We agree that ease-of-use and standards are important: for this reason, our implementation exposes each MAGICAL task and variant as a separate Gym environment, which should be easy to integrate with existing code. However, beyond the API, we do not think that small adjustments to existing benchmarks would suffice to evaluate generalisation. For instance, Section 5 notes that while it's possible to create "test" variants of Gym MuJoCo environments (e.g. the disabled ant), the underlying goal is still very simple, which limits how many distinct test variants can be created (there are only so many ways to run in one direction). In contrast, the object manipulation tasks in the MAGICAL benchmark have more complex goals that allow us to test many different axes of variation. We believe this justifies the introduction of a new benchmark, particularly given how few IL-specific benchmarks there are today.

Table AR1: Preliminary results for new methods on a subset of tasks (three seeds, 500k time steps per run).

| | Demo | Jitter | Layout | Colour | Shape | CountPlus | Dynamics | All |
|---|---|---|---|---|---|---|---|---|
| **MoveToCorner** | | | | | | | | |
| GAIL | 0.43±0.37 | 0.36±0.31 | - | 0.35±0.32 | 0.42±0.38 | - | 0.44±0.36 | 0.30±0.20 |
| WGAIL-GP | 0.04±0.03 | 0.05±0.05 | - | 0.01±0.01 | 0.08±0.10 | - | 0.05±0.05 | 0.01±0.01 |
| Apprenticeship learning | 0.00±0.00 | 0.00±0.00 | - | 0.00±0.00 | 0.00±0.00 | - | 0.00±0.00 | 0.00±0.00 |
| **MoveToRegion** | | | | | | | | |
| GAIL | 1.00±0.00 | 1.00±0.00 | 0.54±0.14 | 0.45±0.20 | - | - | 1.00±0.00 | 0.26±0.08 |
| WGAIL-GP | 0.97±0.01 | 0.97±0.01 | 0.47±0.02 | 0.25±0.02 | - | - | 0.97±0.02 | 0.14±0.01 |
| Apprenticeship learning | 0.33±0.47 | 0.33±0.46 | 0.10±0.14 | 0.09±0.12 | - | - | 0.33±0.47 | 0.05±0.07 |
| **FixColour** | | | | | | | | |
| GAIL | 1.00±0.00 | 0.67±0.11 | 0.21±0.03 | 0.32±0.03 | 1.00±0.00 | 0.16±0.04 | 0.97±0.04 | 0.14±0.04 |
| WGAIL-GP | 0.65±0.42 | 0.31±0.21 | 0.06±0.02 | 0.10±0.04 | 0.66±0.40 | 0.02±0.03 | 0.66±0.37 | 0.04±0.03 |
| Apprentice. learn. | 0.01±0.01 | 0.00±0.00 | 0.01±0.01 | 0.05±0.02 | 0.03±0.02 | 0.00±0.00 | 0.01±0.02 | 0.01±0.00 |

[Meta-Review · NeurIPS 2020]

The rebuttal managed to successfully address some major concerns and to address some misunderstandings. A very nice paper proposing an interesting new benchmark. R4 helped R2 write the review, we added an additional reviewer such that R4 could directly participate in the discussion (hence that score only counts once).